# A Conserved Acidic Residue in the C-Terminal Flexible Loop of HIV-1 Nef Contributes to the Activity of SERINC5 and CD4 Downregulation

**DOI:** 10.3390/v15030652

**Published:** 2023-02-28

**Authors:** Claudia Firrito, Cinzia Bertelli, Annachiara Rosa, Ajit Chande, Swetha Ananth, Hannah van Dijk, Oliver T. Fackler, Charlotte Stoneham, Rajendra Singh, John Guatelli, Massimo Pizzato

**Affiliations:** 1Department of Cellular, Computational and integrative Biology, University of Trento, 38123 Trento, Italy; 2Department of Infectious Diseases, Integrative Virology, University Hospital Heidelberg, 69120 Heidelberg, Germany; 3German Center for Infection Research (DZIF), 69120 Heidelberg, Germany; 4Department of Medicine, University of California San Diego, La Jolla, CA 92093, USA; 5VA San Diego Healthcare System, San Diego, CA 92161, USA

**Keywords:** HIV-1, Nef, SERINC5

## Abstract

The host transmembrane protein SERINC5 is incorporated into retrovirus particles and inhibits HIV-1 infectivity. The lentiviral Nef protein counteracts SERINC5 by downregulating it from the cell surface and preventing its incorporation into virions. The ability of Nef to antagonize the host factor varies in magnitude between different HIV-1 isolates. After having identified a subtype H *nef* allele unable to promote HIV-1 infectivity in the presence of SERINC5, we investigated the molecular determinants responsible for the defective counteraction of the host factor. Chimeric molecules with a subtype C Nef highly active against SERINC5 were constructed to locate Nef residues crucial for the activity against SERINC5. An Asn at the base of the C-terminal loop of the defective *nef* allele was found in place of a highly conserved acidic residue (D/E 150). The conversion of Asn to Asp restored the ability of the defective Nef to downregulate SERINC5 and promote HIV-1 infectivity. The substitution was also found to be crucial for the ability of Nef to downregulate CD4, but not for Nef activities that do not rely on the internalization of receptors from the cell surface, suggesting a general implication in promoting clathrin-mediated endocytosis. Accordingly, bimolecular fluorescence complementation revealed that the conserved acidic residue contributes to the recruitment of AP2 by Nef. Altogether, our results confirm that Nef downregulates SERINC5 and CD4 by engaging a similar machinery and indicates that, in addition to the di-leucine motif, other residues in the C-terminal flexible loop are important for the ability of the protein to sustain clathrin-mediated endocytosis.

## 1. Introduction

All primate lentiviruses express Nef, a myristoylated protein translated abundantly from a doubly spliced RNA from the early stages of an infection. While not absolutely essential for virus replication, Nef is a pathogenic factor crucially contributing to disease progression, as documented in patients and animals infected with HIV and SIV, respectively [1,2,3]. Nef is a multifunctional protein with the ability to modulate cell signaling in T-cells and macrophages [4], and to alter the abundance of diverse cell surface proteins. Such multifunctionality parallels the ability of Nef to interact with a plethora of host factors and to induce the internalization of different cell surface molecules [5]. Nef is thought to function as an adaptor capable of bridging cellular cargoes to the vesicular transport system. Pivotal to its activity of downregulation of cell surface proteins is the interaction with clathrin adaptor complexes (AP1, AP2 and AP3) required for transporting, among other proteins, MHC-I complexes, the viral entry receptor CD4, and the infectivity inhibitor SERINC5 into the intracellular endolysosomal compartment [6]. The mechanisms of downregulation of MHC-I and CD4 are the best characterized so far. Nef is thought to target MHC-I molecules located at the TGN and at the cell surface to the lysosomal degradative compartment following the formation of a ternary complex with AP1, involving the acidic and poly-proline clusters located on the most N-terminal third of the molecule (62EEEE65 and 72PxxP75) [7]. Cell surface CD4 is internalized into clathrin-coated pits and then targeted to the lysosomal compartment following the formation of a complex with AP2 involving the C-terminus proximal di-leucine-based sorting signal (164LL165) of Nef [8]. The molecular determinants governing CD4 downregulation also include the N-terminal motif 57WL58 of Nef, perhaps implicated in the direct interaction between the two proteins, since it was found to associate with a peptide derived from the CD4 cytoplasmic tail [8].

SERINC5 is a multipass transmembrane protein found to inhibit the infectivity of retrovirus particles [9,10] by impairing the translocation of the viral core into the cytoplasm [11,12]. While the ability of Nef to target SERINC5 is highly conserved across alleles derived from all primate lentiviruses [13], the mechanism of downregulation remains to be fully understood, although it appears to share features with the mechanism of CD4 downregulation [14]. The removal of SERINC5 from the cell membrane requires clathrin-mediated endocytosis and depends on the same molecular Nef determinants that are currently known to be crucial for CD4 downregulation, including the di-leucine-based sorting signal (164LL165) and the conserved 57WL58 [15]. However, a direct interaction between the two proteins, though suggested by bimolecular fluorescence complementation, remains to be formally demonstrated [14].

Virus evolution in vivo can result in naturally occurring polymorphisms that have sometimes proven useful to investigate the molecular determinants functionally important for viral activities. For example, the selective pressure dictated by immunological constraints can force the virus to select changes which allow the escape from immune recognition at the expense of the efficiency of some of its functions. This was shown, for example, with *nef* alleles from viruses isolated from elite controllers, which contain CTL escape mutations partially compromising the ability to counteract SERINC5 [16].

Having observed significant variations of the magnitude of the Nef counteraction of SERINC5, we sought to investigate the nature of such variability and further map the determinants of Nef that are important for its activity in targeting the host factor.

## 2. Materials and Methods

### 2.1. Plasmids

Env- and Nef-defective HIV-1 NL4-3–derived plasmid contains frameshift mutations to inactivate the Nef and Env ORF, and was previously described [17]. Single round Env-defective viruses were complemented with a PBJ5 plasmid expressing HIV-1 HXB2 env, as described previously [18].

PBJ5 plasmids encoding *nef* genes from SF2, YU2, JRFL (subtype B), 97ZA012 (subtype C), 94UG114 (subtype D), 93BR020 (subtype F), and 90CF056 (subtype H), with a C-terminal HA tag, were described elsewhere [17]. DNA encoding Nef^JRFL^ (corresponding to the sequence with accession number U63632.1) was custom synthesized (Thermofisher Scientific, Waltham, MA, USA) and subcloned into a pBJ5 plasmid with a HA tag at the C-terminus. Nef^Fs^, used as a negative control, was generated mutagenizing the Nef^LAI^ sequence to introduce a frameshift at the unique XhoI site, which disrupts the ORF of Nef starting from codon 34. All *nef* alleles and mutants were also subcloned into pIRES2-GFP for FACS analyses.

Single point Nef mutants (V134E, Y139F, D150N, N150D) were generated by site directed mutagenesis using the quick-change method and the high fidelity PFU polymerase (Promega, Madison, WI USA). DNAs encoding chimeric Nef proteins (CH65, CH130, HC65, HC130, CH152, CH173, CH183, CH194, CHC131-151, HCH 131-151) were generated using an overlap extension PCR method and inserted into pBJ5 (In-fusion cloning technology, Takara, Saint-Germain-en-Laye, France). PCR-derived mutants were sequenced to confirm their accuracy.

Plasmids encoding the AP-2 sigma (σ2) or alpha (α) subunits fused to the Venus-C-terminal fragment were a gift from Thomas Smithgall [19]. Plasmids encoding the various Nef proteins fused to the Venus-N-terminal fragment were generated by inserting the ORF of the different Nef variants into the plasmid PCDNA-Venus.

The SERINC5-HA used for infectivity experiments was expressed from pBJ6 plasmid [9]. PBJ5-SERINC5-iFlag-HA was generated by inserting an internal FLAG-tag in the fourth extracellular loop (between 290–291 aa) of PBJ5-SERINC5 [20]. 

### 2.2. Cells Lines

HEK293T and HEK293 cells were grown in Dulbecco’s Modified Eagle’s Medium (DMEM) supplemented with 10% FCS plus 2 mM glutamine, 100 U/mL penicillin, and 100 µg/mL streptomycin.

TZM-zsGreen [9] are derived from TZM-bl indicator cells (from the NIH AIDS Reagent Program), and were maintained in Dulbecco’s Modified Eagle’s Medium (DMEM) supplemented with 10% FCS, 2 mM glutamine, 100 U/mL penicillin, and 100 µg/mL streptomycin.

Jurkat TAg (JTAg) cells were generated from Jurkat E6.1 T-cells by the stable overexpression of SV40 Large T-antigen. The MT4 T-cell line was obtained from the NIH AIDS Reagent Program. JTAg and MT4 were grown in suspension in RPMI supplemented with 10% FCS, 2 mM L-Glutamine, 100 U/mL penicillin, and 100 µg/mL streptomycin.

All cell lines were maintained in a humidified incubator at 37 °C and 5% CO_2_. The cultures were tested and resulted mycoplasma-free.

### 2.3. Viruses and Infectivity Assay

HIV-1 virions restricted to a single round of replication were produced by transfecting HEK293T cells with the calcium phosphate precipitation method. One day before transfection, HEK293T cells were seeded in 6-well plates at a density of 300,000 cells/well. Cells were co-transfected with 2.5 μg of NL4-3–based plasmid, 0.5 μg of env-expressing plasmid, 1 μg of Nef-expressing plasmid, or an equivalent amount of control vector, 1 μg of PBJ6-SERINC5- HA, or an equivalent amount of control vector. In the absence of exogenously expressed SERINC5, Nef has a negligible effect on the infectivity of the virus produced from these cells, since they have a reduced expression of endogenous SERINC5 following disruption of the gene with CRISPR-Cas9 [21]. Virus-containing supernatants were collected 48 h post-transfection, clarified by centrifugation at 300× *g* for 5 min, and passed through filters with 0.45-μm pores. Before infection assays, viruses (prepared in quadruplicate) were quantified using the SG-PERT reverse transcription assay [9], diluted fivefold in a series of six steps, and used to infect TZM-bl-zsGreen reporter cells seeded one day before infection in 96-well plates. TZM-bl-zsGreen is a modified version of TZM-bl containing an integrated nlszsGreen reporter gene under the transcriptional control of the HIV-1 long terminal repeat. The infection of reporter cells was scored using the Operetta High Content Imaging System (Perkin Elmer, Waltham, MA, United States) after counterstaining nuclei with Hoechst 33342 for each virus dilution. Those values falling into a linear dilution range (normally below 20% of infected cells) were used to calculate infectivity. Infectivity was calculated by dividing the number of infected cells in a well for the amount of reverse transcriptase activity associated to the virus inoculum, measured in mU. When indicated, results were expressed as a percentage of an internal control sample. 

### 2.4. FACS Analyses

The effects of Nef on cell surface expression levels of CD4, MHC-I and SERINC5 were measured using MT4 cells (CD4) or the JTAg cell line (MHC-I and SERINC5). To analyze SERINC5 surface levels, a modified SERINC5 harboring an internal FLAG epitope within the fourth extracellular loop (SERINC5-iFLAG-HA) was overexpressed from the PBJ5 plasmid. To facilitate the identification of transfected cells, the indicated *nef* alleles were expressed from a pIRES2-eGFP plasmid. For each sample, 10 × 10^6^ JTAg cells were electroporated with 5 µg of PBJ5- SERINC5-iFLAG-HA and 20 µg of pIRES2-Nef-eGFP. Instead, in case of CD4 and MHC-I, the surface downregulation of the endogenous molecules was monitored; hence, MT4 and JTAg cells were electroporated with 20 µg of pIRES2-Nef-eGFP and, as carrier DNA, 5 µg of pBluescript plasmid. For all electroporations, the Biorad Gene Pulser Xcell was used, choosing the pre-set protocol for Jurkat cells. Forty-eight hours post electroporation, cells were stained with mouse anti-CD4 (1:1000), mouse anti-HLA-A,-B,-C (1:1000) or mouse anti-FLAG (1:500). PBA buffer (PBS, 5% BSA, 0.1% sodium azide) was used for both the washing and the antibodies incubation steps. As a secondary antibody, an APC-conjugated goat anti-mouse IgG (1:500) was used. Samples were acquired and analyzed with the BD Canto flow cytometer. Post-acquisition analysis was performed with FlowJo v10 software (BD Life Sciences, Franklin Lakes, NJ, USA). A viable population of GFP^+^ single cells was identified sequentially based on FSC-A vs SSC-A, FSC-A vs FSC-H and FITC vs APC. The flow cytometry histograms represent the APC fluorescence intensity of the indicated samples measured in the GFP^+^ population. 

Mouse anti-FLAG (clone M2) was purchased from Sigma-Aldrich, while mouse anti-CD4 (clone RPA-T4) and mouse anti-HLA-A,-B,-C (clone G46-2.6) were obtained from Becton Dickson. APC-conjugated goat anti-mouse IgG was obtained from Jackson Immunoresearch.

### 2.5. Western Blotting

Proteins from cell lysates were resolved through 11% SDS-PAGE and analyzed by western blotting. In brief, cells were harvested 48 h after transfection and centrifuged at 300× *g*. Cell were then resuspended in a RIPA lysis buffer and incubated on a rolling wheel for 30 min at 4 °C. Clarified supernatants were then mixed with a 2X Laemmli buffer. Samples were boiled at 95 °C for 5 min before being loaded onto 11% acrylamide, and then transferred on an Immobilon FL-PVDF membrane (Millipore, Burlington, MA, USA). The membranes were blocked in Odyssey Blocking Buffer (Li-COR, Lincoln, NE, USA) diluted 1:1 in TBS for 30 min at room temperature. Membrane probing was performed with a mouse anti-HA antibody (HA.11, clone 16B12; Biolegend, San Diego, CA, USA), a mouse or rabbit β-actin antibody (Li-COR); secondary antibodies used were goat anti-rabbit IRDye 800 and goat anti-mouse IRDye 680 (Li-COR). Proteins bands were visualized using the Li-COR Odyssey infrared imaging System (Li-COR).

### 2.6. Lck Retargeting Assay

Retargeting of Lck in T cells was evaluated by fluorescence microscopy as described [22]. Briefly, 10 × 10^6^ JTAg T-cells were co-transfected with 15 µg eGFP.N1 or 30 µg Nef.eGFP.N1 plasmid DNA and 15 µg Lck.RFP.N1 DNA using a Bio-Rad GenePulser (BioRad, Hercules, CA, USA) at 950 µF, 250 V. Cells were harvested at 24 h post transfection and seeded on poly-L-lysine coated coverslips for 10 min followed by fixation with 3% paraformaldehyde for 10 min. Coverslips were mounted on glass slides using Mowiol. Single Z-plane images were acquired using a Leica SP8 confocal microscope with a 63× objective. Saved .tif files were cropped and adjusted for brightness and contrast (same threshold for all images) using ImageJ software, Version 1.53t (NIH, Bethesda, MD, USA).

### 2.7. BiFC

Bimolecular fluorescence complementation assays were performed in HEK293 cells. HEK293 cells seeded in 12-well plates were transfected using lipofectamine 2000, following the manufacturer’s instructions. Cells were transfected with a total of 1.6 µg plasmid DNA; with equal amounts of AP2α-Venus-C-V5, AP2σ2-Venus-C-V5, Nef-HA-Venus-N and empty control vector plasmid (pcDNA3.1) (400 ng plasmid each). As controls, HEK293 cells were transfected to express single Nef (Venus N) plasmids, or combined AP2α-Venus-C-V5 and AP2σ2-Venus-C-V5. The following day, the cells were removed from the wells by gentle pipetting and pelleted at 300 g for 5 min at 4 °C. The cell pellets were resuspended in 1X PBS to remove residual DMEM, and pelleted again at 300 g for 5 min at 4 °C. The cells were fixed for 15 min in 2% paraformaldehyde (PFA) at 4 °C before analysis using a Novocyte (Agilent, Santa Clara, CA, USA) benchtop flow cytometer. Live single cells were gated using FSC-A v. FSC-H and FSC-A vs SSC-A. The BiFC signal was detected using the FITC channel. 

To analyze the cellular expression of AP2α-Venus-C-V5 and AP2σ2-Venus-C-V5, or single Nef-Venus-N-HA mutants, HEK293 cells were transfected with the equivalent concentrations of each plasmid (400 ng), using the same protocol as above. Cells were harvested for western blot 24 h post-transfection by gentle pipetting and centrifugation at 300× *g* for 5 min at 4 °C. Cell pellets were resuspended in extraction buffer (0.5% Triton X-100, 150 mM NaCl, 25 mM KCl, 25 mM Tris, pH 7.4, 1 mM EDTA) supplemented with a protease inhibitor mixture (Roche Applied Science), and incubated on ice for 30 min, vortexing occasionally. Nuclei were pelleted by centrifugation at 5000× *g* for 10 min, 4 °C, and clarified supernatants were combined with a 2× Laemmli buffer. Samples were boiled at 95 °C for 5 min before being loaded on 12% acrylamide, alongside PageRulerPlus protein standard (Thermo Scientific, Waltham, MA, USA), and then transferred to the Immobilon FL-PVDF membrane (Millipore, Burlington, MA, USA). Membranes were blocked with 5% non-fat milk (BioRad, Hercules, CA, USA) in PBS containing 0.2%Tween-20, for 30 min at room temperature. Membranes were probed using mouse anti-HA antibody 1:1000 (HA.11, clone 16B12; BioLegend, San Diego, CA, USA), mouse anti-V5 antibody 1:1000 (Thermo Invitrogen), mouse anti-GAPDH 1:3000 (GeneTex, Irvine, CA, USA) in PBST containing 1% non-fat milk and 0.05% Sodium azide. Primary antibodies were probed using horseradish peroxidase (HRP)-conjugated goat anti-mouse IgG secondary antibody (BioRad) diluted 1:3000. Chemiluminescence was detected using Western Clarity detection reagent (BioRad) and imaged using a BioRad Chemi Doc imager with BioRad Image Lab v5.1 software.

## 3. Results

### 3.1. The Ability of Nef Proteins to Counteract SERINC5 Varies in Different HIV-1 Isolates

The natural variability of the SERINC5-counteracting activity in HIV-1 Nef proteins was studied by investigating the ability of Nef, derived from different isolates, to rescue the viral infectivity in the presence of the host factor. HIV-1 particles restricted to a single cycle of replication were produced in HEK293T cells cotransfected to express SERINC5 and different *nef* alleles fused to an HA tag at the C-terminus. As shown in Figure 1A, in which the infectivity of the progeny virus was measured on TZM-bl-zsGreen reporter cells, and the ability of Nef proteins to promote HIV-1 infectivity in the presence of SERINC5 was highly variable. Nef derived from HIV-1^YU2^ and from HIV-1^90CF056^, the latter being an isolate belonging to clade H (here called Nef^H^), displayed the weakest activity against the restriction factor, while Nef from HIV-1^SF2^ and isolates belonging to genetic clade C (HIV-1^97ZA012^, here called Nef^C^) and clade F (HIV-1^93BR020^) displayed the strongest activity. To explore whether the different activity of the Nef proteins could relate to a different expression in transfected cells, producer cells were lysed, and the steady-state protein abundance was assessed by detecting the HA tag by immunoblotting (Figure 1B). All alleles appeared to produce a similar amount of protein in lysates of virus-producing cells, except for Nef^YU2^, which was repeatedly detected at a lower level in independent experiments (not shown). Given the defective expression compared to other variants, Nef^YU2^ was excluded from subsequent experiments.

We further investigated whether the counteracting activity of the various Nef proteins against the effect of SERINC5 on infectivity reflects their ability to target the cell surface level of the host restriction factor. Transfected Jurkat TAg cells were therefore tested to investigate the ability of the Nef proteins to downregulate SERINC5 from the cell surface using flow cytometry. For this purpose, the Nef-coding sequences were inserted into a vector upstream of an IRES2-eGFP cassette and transfected together with a plasmid expressing human SERINC5 modified to expose a FLAG tag on the fourth extracellular loop (SERINC5-iFLAG), conveniently detectable with an anti-FLAG antibody. The simultaneous expression of eGFP and Nef from the same RNA allows for the efficient gating of fluorescent cells expressing the viral factor. As shown in Figure 1C and Appendix A, all Nef proteins analyzed could decrease the surface expression level of SERINC5, and the magnitude of this activity mirrored the ability to rescue the infectivity of HIV-1, with Nef^SF2^, Nef^C^ and Nef from a clade F isolate (93BR020) being the most active in downregulating SERINC5. In contrast, Nef^H^ appeared to have the weakest activity. Overall, the extent of SERINC5 downregulation was found to correlate with the ability of Nef to promote HIV-1 infectivity, following a logarithmic function, as shown in Figure 1D, suggesting that a minimal threshold amount of SERINC5 surface expression is sufficient to inhibit virus infectivity. The downregulation of the restriction factor by Nef is therefore required to reach such a threshold and to promote virion infectivity.

### 3.2. A Region at the C-Terminus of Nef and Upstream the Leucine-Based Motif Contributes to the Activity against SERINC5

The sequence of Nef^C^, which has a strong activity against SERINC5 (Nef^C^, Figure 2A), shares 76% identity with the sequence of Nef^H^, the weakest allele against the host factor. In the attempt to identify the molecular determinants responsible for the high variability of the protein activity against SERINC5, chimeric Nef molecules were constructed by producing proteins which combine Nef^C^ and Nef^H^ regions, as illustrated in Figure 2A. Chimeric molecules were initially designed by dividing the molecule into three equal parts, with swaps at amino acid 65 (located in the N-terminal arm of the protein) and 130 (within the folded core domain of Nef). All chimeric molecules appeared to be equally well expressed in the lysates of virus-producing cells, indicating that the molecular swaps are well-tolerated and do not affect the steady state expression of the proteins (Figure 2B). In infectivity assays, the higher ability of Nef^C^ to counteract SERINC5 segregated with the region downstream of residue 130, i.e., the C-terminal third of the molecule (Figure 2C). This region includes the flexible loop, which contains the sorting signal important for the interaction with the clathrin adaptor complexes.

### 3.3. A 21 aa Stretch of Nef Upstream the Di-Leucine Motif Is Crucial for the Activity against SERINC5

To narrow down the sequence responsible for the high activity of Nef^C^ against SERINC5, further swaps with Nef^H^ were performed within the C-terminus of the molecule. The C-terminal portion of Nef^C^ was further divided into four parts (130-151 aa; 152-173 aa; 173-194 aa; 194-207 aa), and chimeric Nef proteins were generated by fusing increasingly smaller portions of the C-terminus from Nef^H^ (Figure 3A). While the new chimeric proteins appeared to be equally expressed in virus-producing cells (Figure 3B), their anti-SERINC5 activity on infectivity was progressively restored by replacing the C-terminus with the corresponding regions from Nef^C^ (Figure 3C). The Nef^C^ region between residues 130 and 152 associated with a discrete step in the increase of activity on infectivity.

To confirm the importance of this region, the 130-151 amino acidic stretch was swapped reciprocally between Nef^C^ and Nef^H^. The modification did not impact the stability and expression level of the proteins as shown by immunoblot (Figure 3D). However, while the replacement of the 21 amino acids of Clade C Nef with those derived from Clade H diminished the activity of Nef^C^ against SERINC5 two-fold, the reciprocal swap produced a full rescue of the activity of Nef^H^ (Figure 3E). This result indicates that the 130-151 region of Nef is functionally crucial, and in the context of a weaker Nef molecule represents a determinant which modulates the activity against SERINC5.

### 3.4. Residue 150 Is Crucial for the Activity of Nef Clade H on Infectivity

The Nef^C^ and Nef^H^ regions encompassing residues 130 to 151 diverge in only three positions: 134, 136, and 150 (Figure 4A). Of note, at position 150, an acidic amino acid is present in all Nef proteins analyzed, with the only exception of Nef^H^, which features an Asn. In contrast, different amino acids are present at positions 134 and 136 in different Nef proteins. To investigate whether the gain-of-function transferred to Nef^H^ from Nef^C^ can be attributed to a single amino acid, the cDNA encoding the chimeric protein (HCH) was mutated to singularly restore the three amino acids present in Nef^H^. The effect of the three single substitutions (V134E, Y136F and D150N) on the activity against SERINC5 was tested in a single-cycle infectivity assay. The mutations did not negatively affect the protein expression in virus-producing cells, since all proteins appeared to maintain a similar steady-state expression level (Figure 4B). While two of the three mutants (V134E and Y136F) did not show any substantial difference compared to the parental protein, the D150N substitution was associated with a steep reduction of activity on infectivity compared to the unchanged protein (Figure 4C).

To further test the importance of Nef residue 150, we reciprocally swapped the single amino acid in the context of Nef^C^ and Nef^H^ proteins. The mutants, similarly expressed in the lysate of virus-producing cells (Figure 4D), were tested for their effect on infectivity along with their parental controls. As shown in Figure 4E, the conversion of Asn to Asp (N150D) restored the activity of Nef^H^ on the infectivity to the level Nef^C^. In contrast, the reciprocal mutation inserted into Nef^C^ (D150N) resulted in only a weak decrease of the effect of the protein on HIV-1 infectivity (Figure 4E), indicating that the nature of residue 150 plays a more critical role in the context of Nef^H^ compared to Nef^C^.

An analysis of the Los Alamos database reveals that 96.09% of available HIV-1 Nef sequences (6487 sequences) displays either a Glu or Asp in position 150 (Figure 4F). The presence of an acidic residue appears similarly prevalent in this position of Nef proteins from different genetic subtypes, indicating a general constraint.

### 3.5. Residue 150 Is Crucial for the Nef Downregulation Activity of SERINC5 and CD4

Having documented the effect on infectivity, the role of Nef residue 150 for SERINC5 downregulation from the cell surface was tested. Reflecting its ability to support virus infectivity, the ability of Nef^H^ to downregulate SERINC5 was restored to a level comparable to that of Nef^C^ by introducing the N150D substitution (Figure 5A and Appendix A). However, the reciprocal substitution in the context of Nef^C^ only had a subtle effect on SERINC5 surface expression levels, correlating with the smaller effect of this substitution on virus infectivity (Figure 4E).

Given that the downregulation of SERINC5 and CD4 by Nef are thought to be mechanistically linked, the importance of residue 150 for the ability of Nef to reduce CD4 surface expression was also investigated. Compared to Nef^C^, Nef^H^ displayed a reduced ability to downregulate CD4 (Figure 5B). Replacing Asn with Asp rescued the CD4 downregulation activity of Nef^H^ to a level similar to that of Nef^C^. Mirroring the effect on infectivity, the reciprocal conversion of the amino acid in Nef^C^ (D150N) caused only a modest alteration on the activity on CD4 downregulation (Figure 5B). Of note, the presence of an Asn at position 150 in Nef^H^ had an effect on CD4 and SERINC5 downregulation similar to the disruptive effect observed with mutating the di-leucine sorting signal in Nef^LAI^ (Figure 5A,B). Moreover, while the effect of the D150N substitution impaired the ability of Nef^C^ to promote virus infectivity by 50% (Figure 4E), it did not have an evident effect on the SERINC5 surface expression level measured by FACS (Figure 5A). This discrepancy could be attributable to a minimal alteration of SERINC5 expression not detected by flow cytometry. Indeed, we have documented that SERINC5 is a very potent inhibitor of virions carrying the HXB2 envelope glycoprotein [9], and even small variations of its expression levels could result in noticeable effects on infectivity.

In contrast, Nef activities that do not involve the c-terminal flexible loop of Nef but are mediated by the PxxP motif [23,24] were unaffected by the variation of the residue at position 150: both the cell surface downmodulation of MHC-I (Figure 5A) and the retargeting of the peripheral membrane protein Lck from the plasma membrane to an intracellular compartment (Figure 5B) remained mostly unaltered irrespectively of the *nef* allele and the amino acid change. This analysis also revealed that variations at position 150 did not significantly impact the subcellular localization of Nef (Figure 5B, left). Altogether, these data demonstrate that residue 150 is equally important for the activity of Nef against SERINC5 and CD4, confirming the mechanistic correlation between the downregulation activity of both host molecules.

### 3.6. Residue 150 Is Important for Nef-AP2 Interaction

The ability of Nef to downregulate SERINC5, as for CD4, depends on the clathrin adaptor complex AP2 [25]. Accordingly, the AP2 binding motif (ExxxLL) located on the C-terminal flexible loop of the protein is required for the downregulation of both cell surface proteins by Nef [9,10,26]. Residue 150 is located at the base of the C-terminal flexible loop that also contains the AP2 binding motif. We speculate that variations at this residue could perturb the interaction of Nef with the clathrin adaptor complex. To investigate this possibility, the interaction between the two binding partners was tested using BiFC by fusing the proteins to complementary non-fluorescent portions of the yellow fluorescent protein Venus. When the protein partners interact, the non-fluorescent fragments restore a bright fluorescent signal, which is a measure of the level of the interaction. To this end, different Nef variants were fused to the N-terminal fragment of Venus (VN), whereas the α and σ2 subunits of AP2, which together with β2 and μ2 assemble the clathrin adaptor complex, were fused to the C-terminal fragment (VC). To facilitate detection by immunoblotting, each protein was tagged with V5 (for AP2 α and σ2) or HA (for Nef) epitope tags. Fluorescence complementation resulting from the interaction between Nef and AP-2 α and σ2 subunits was detected by flow cytometry (Figure 6A and Appendix A for a representative FACS histogram). The fluorescent signal required the expression of both binding partners, since a weak background signal was detected when only one partner was expressed (see Figure 6A and Appendix A). As a control we used the Nef protein from the LAI isolate (Nef^LAI^) and its variant lacking the leucine-based sorting motif (NefLL/AA), which has impaired AP2 binding. This mutant shows a 50% decrease of fluorescence complementation, confirming that the Nef-AP2 interaction crucially contributes to the signal (Figure 6A and Appendix A). As shown in Figure 6B, all binding partners were equally expressed in the lysates of the transfected HEK293 cells.

The different *nef* alleles co-expressed with the AP2 subunits generated different levels of fluorescence. Nef^C^ and Nef^LAI^ generated the brightest signal, and Nef^H^ produced the lowest level of fluorescence, suggesting that Nef^H^ has an overall lower ability to engage with AP2 (Figure 6A left, results of three independent experiments).

The role of residue 150 in the context of Nef^H^ and Nef^C^ was therefore investigated. Overall, the N150D substitution in Nef^H^ was associated with a 50% higher fluorescence level compared to the WT counterpart (Figure 6B right), indicating an increased ability to recruit AP2 by the mutant protein and correlating with increased SERINC5 and CD4 downregulating activity. In contrast, the reciprocal substitution at residue 150 in the context of Nef^C^ (D150N) had the opposite effect, resulting in a 50% decrease of fluorescence and confirming that the acidic amino acid in position 150 contributes to the interaction with AP2. Of note, while the D150N substitution in Nef^C^ associated with a decreased signal compared to the WT protein, the level of fluorescence in three independent experiments remained significantly higher than the fluorescence produced by the WT Nef^H^ protein, and was comparable to the fluorescence generated by the N150D Nef^H^ mutant (see Figure 6A). These data suggest that, while residue 150 affects the efficiency of Nef to recruit AP2 in the context of both Nef^H^ and Nef^C^, the latter maintains a level of binding to AP2 which is sufficient for an effective downregulation of SERINC5 and CD4.

## 4. Discussion

The allelic variability of Nef observed in different isolates has proven instrumental for defining molecular determinants contributing to the ability of the viral protein to counteract SERINC5 [16,27,28,29,30]. Similarly, in this study, a *nef* allele derived from a subtype H isolate with defective SERINC5 downregulation activity has been used to identify an unusual polymorphism in position 150, which affects both SERINC5 counteraction and CD4 downregulation. Restoring an acidic residue in this position, as found in more than 96% of known HIV-1 sequences, allowed for the full recovery of the activity of the defective *nef* allele and pinpointed a novel molecular determinant contributing to the ability of the protein to downregulate both surface molecules.

Residue 150 is located at the N-terminal edge of the Nef flexible loop, which makes contact with a pocket in the σ2 subunit of AP2 [8] and features the di-leucine sorting motif required for both CD4 and SERINC5 downregulation. We hypothesized that residue 150 affects the ability of Nef to recruit AP2, since it is instrumental for the downregulation of both molecules. Using BiFC, we confirmed that a more efficient recruitment of the clathrin adaptor complex by Nef is favored by an Asp in position 150. Of note, while in the context of the weak HIV-1^90CF056^, the Asn to Asp change at 150 was strongly required for SERINC5 and CD4 downregulation; the opposite change, in the context of the stronger HIV-1^97ZA012^, has only a minimal consequence on Nef activity. This suggests that other residues must contribute to the higher ability of the protein to engage with AP2 and compensate for the presence of a suboptimal amino acid in position 150. These results confirm the indications by Ren et al. [31] that several determinants, in addition to the di-leucine sorting motif, contribute to the ability of Nef to interact with cellular adaptins. Accordingly, Nef was reported to form an extensive interaction with the sigma subunit of AP2, directly involving different residues of the C-terminal loop, including the amino acids intervening within the dileucine motif [30]. Other residues, such as the acidic dipeptide at 174, can instead indirectly contribute to AP2 binding by stabilizing the conformation in relation to the Nef globular core. Available structural data of the Nef-AP2 complex [31] and the recently reported trimeric Nef-AP2-CD4 complex do not support evidence for a direct interaction of residue 150 with AP2 [8]. It therefore appears more plausible that a negatively charged residue stabilizes the molecular structure of the loop, with direct consequences for both CD4 and SERINC5 downregulation by Nef. Future structural studies would be required to elucidate such a possibility by visualizing the conformation of the mutant viral protein. Of note, in addition to an effect on AP2 binding, the possibility exists that a polymorphism at residue 150 would also affect the binding of β-COP to the EE154-155 motif, thus altering the ability of Nef to target surface molecules to degradation in the lysosome compartment [32]. This possibility remains to be explored.

The recurrence of a Glu or Asp at 150 in most *nef* alleles demonstrates the general constraint for an acidic amino acid in this position, with a different residue found only in 3.9% of the alleles annotated in the Los Alamos database. This could indicate that in some circumstances an alternative selective pressure favors the fixation of a different amino acid. HIV-1 Nef is a protein expressed at high levels soon after infection, and is one of the viral proteins that originate most CTL epitopes. A few of these epitopes map on the C-terminal flexible loop and overlap with residue 150 [33,34,35,36,37,38,39,40]. Though no escape variants affecting this position have been reported yet, it remains possible that the pressure required to escape CTL recognition or to prevent peptide processing could contribute to such polymorphism, as already documented for other regions of Nef that are important for SERINC5 downregulation [16].

In conclusion, though recent reports have identified Nef polymorphisms which uncouple SERINC5 antagonism and CD4 downregulation, our results confirm that the engagement of Nef with clathrin adaptor complexes is a unifying requirement for the ability of the viral protein to downregulate the two membrane proteins and, eventually, to sort them into internal cellular compartments, such as the late endosomes.

## Figures and Tables

**Figure 1 viruses-15-00652-f001:**
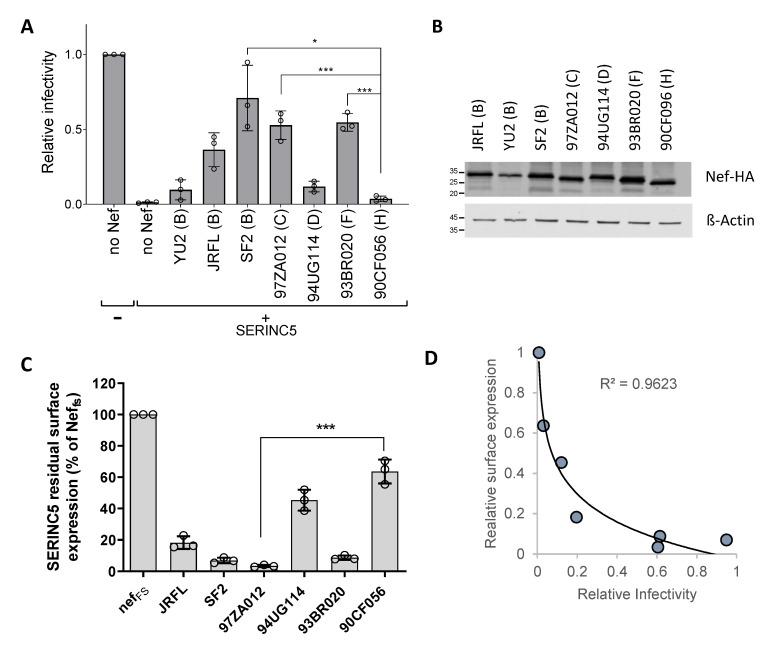
The ability of Nef to counteract SERINC5 is variable across the different viral isolates. (**A**,**B**) Nef proteins from isolates derived from different genetic subtypes have a variable ability to promote HIV-1 infectivity in the presence of SERINC5. (**A**) Infectivity of HIV-1 limited to a single cycle of replication produced in the presence of SERINC5 and the indicated *nef* alleles was determined on TZM-bl-ZsGreen indicator cells after normalization by RT-activity. Bar graphs represent the mean infectivity relative to the infectivity in the absence of SERINC5. Error bars represent s.d. from three biological replicates, unpaired two-tailed *t*-test, * *p* < 0.05, *** *p* < 0.001 (**B**) Detection of HA-tagged Nef proteins expressed in trans in producer cells from (**A**) by western blot. The letters in parentheses indicate the genetic subtype of the viral isolate. (**C**,**D**) The ability of Nef proteins to counteract the inhibition of infectivity by SERINC5 correlates with their ability to downregulate SERINC5 from the cell surface. (**C**) Residual SERINC5-iFLAG cell surface levels in cells co-transfected with the indicated *nef* alleles expressed upstream of an IRES2-eGFP cassette to allow for the exclusive gating of Nef-expressing cells. Nef^Fs^ stands for Nef-frameshift, a control plasmid with the Nef ORF disrupted as explained in Materials and Methods. The residual SERINC5 surface expression was measured as median fluorescence intensity, and values are expressed as a percentage of the Nef^Fs^ control. Error bars represent s.d. from three biological replicates, unpaired two-tailed *t*-test, *** *p* < 0.001 (Graphpad). (**D**) Dot plots correlating the relative infectivity from (**A**) and the residual SERINC5 surface expression from (**C**), with a logarithmic regression curve fitted.

**Figure 2 viruses-15-00652-f002:**
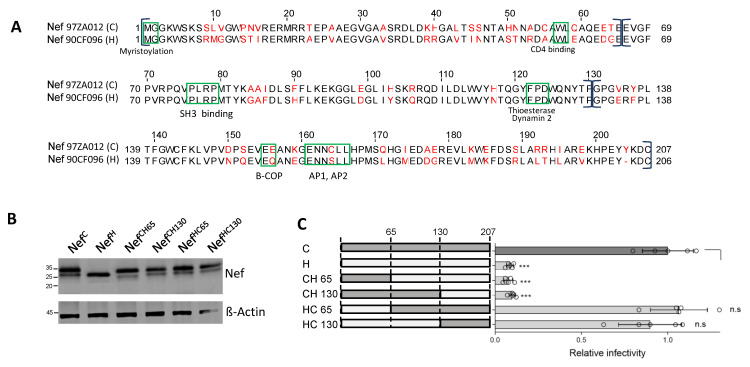
The defective SERINC5 counteracting activity of Nef^H^ maps in the C-terminus third of the molecule. (**A**): Pairwise alignment of Nef^H^ and Nef^C^ showing the divergent residues (in red), some notable functional motifs (green rectangles), and the portions of the molecules swapped for the generation of chimeric proteins (brackets). (**B**): The indicated HA-tagged chimeric Nef proteins are detected by western blot in the lysates of virus producing cells. (**C**): Infectivity of HIV-1 limited to a single cycle of replication produced in the presence of SERINC5 and the indicated Nef proteins. Bar graphs represent the mean infectivity relative to the infectivity of the virus produced with Nef^C^. Mean ± s.d. (Biological quadruplicates), unpaired two-tailed *t*-test, *** *p* < 0.001, (GraphPad).

**Figure 3 viruses-15-00652-f003:**
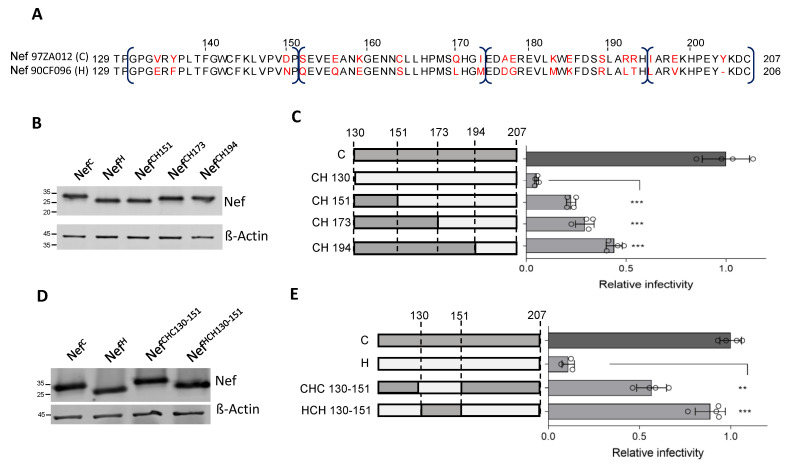
A 21 amino acid sequence from Nef^C^ rescues the SERINC5 counteracting activity of Nef^H^. (**A**): Pairwise alignment of the C-terminus third of Nef^H^ and Nef^C^ showing the divergent residues (in red) and the portions of the molecules swapped for the generation of chimeric proteins within the C-terminal third of Nef (brackets). (**B**,**D**): Western blots of lysates from virus producing cells for the detection of HA-tagged chimeric Nef molecules, as indicated. (**C**,**E**): Infectivity of HIV-1 limited to a single cycle of replication produced in the presence of SERINC5 and the indicated Nef proteins. Bar graphs represent the mean infectivity relative to the infectivity of the virus produced with Nef^C^. Mean ± s.d. (biological quadruplicates), unpaired two-tailed *t*-test, ** *p* <0.01, *** *p* <0.001, (GraphPad).

**Figure 4 viruses-15-00652-f004:**
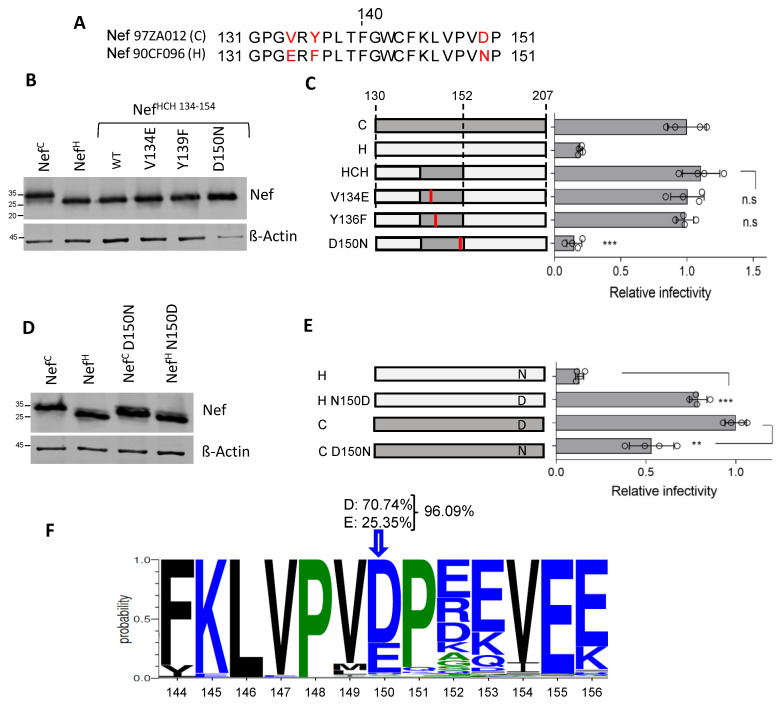
Conversion of Asn to Asp in position 150 fully restores the counteraction activity of Nef^H^. (**A**) Pairwise alignment of the 21 aa minimal sequence which transfers the SERINC5 counteracting activity from Nef^H^ to Nef^C^ , indicating the presence of three divergent residues (in red). (**B**,**C**): Within the minimal sequence rescuing the activity of Nef^H^, D150 is required for SERINC counteraction. (**D**,**E**): D150 is sufficient to restore the SERINC5 counteracting activity in Nef^H^. (**B**,**E**) show western blots of lysates from virus producing cells for the detection of the indicated HA-tagged Nef. (**C**,**E**) show the infectivity of HIV-1 limited to a single cycle of replication produced in the presence of SERINC5 and the indicated Nef proteins, relative to the infectivity measured with Nef^C^. Bar graphs represent the mean relative infectivity. Mean ± s.d. (biological quadruplicates), unpaired two-tailed *t*-test, ** *p* < 0.01, *** *p* < 0.001 (GraphPad). (**F**): Sequence logo representing the conservation of the amino acid residues of a sequence including residue 150, built from HIV-1 6487 sequences in the Los Alamos database (accessed on 28 November 2022).

**Figure 5 viruses-15-00652-f005:**
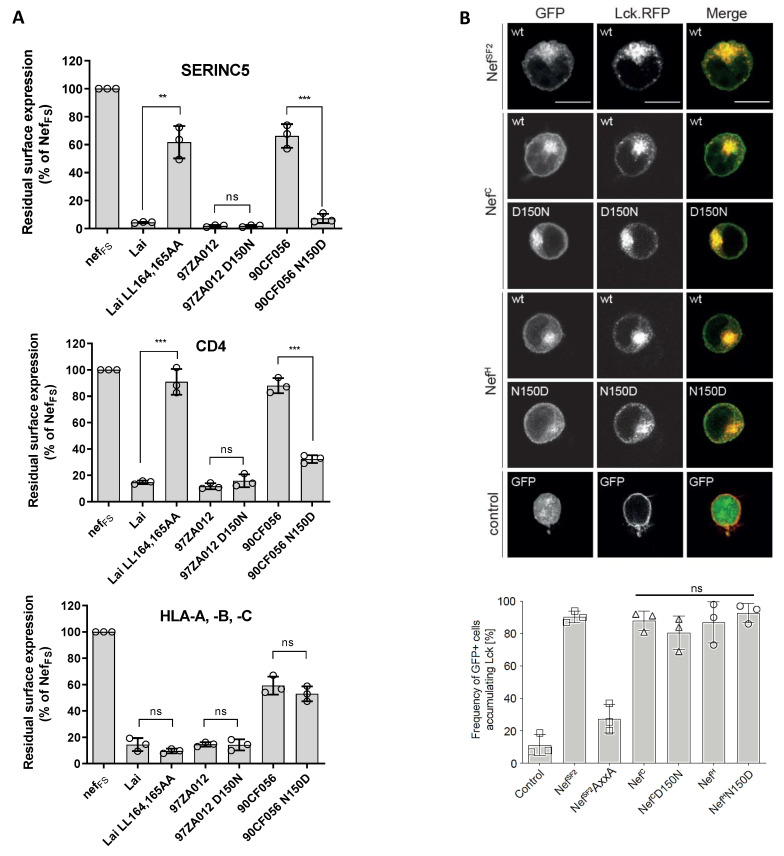
The D150N substitution restores the ability of Nef^H^ to downregulate SERINC5 and CD4. (**A**): the effect of different Nef proteins on the surface expression level of SERINC5, CD4 and MHC-I. Relative cell surface levels in cells co-transfected with the indicated *nef* sequences expressed upstream of an IRES2-eGFP cassette to allow the exclusive gating of Nef-expressing cells. Residual surface expression was measured as median fluorescence intensity, and values are expressed as a percentage of the Nef^Fs^ control. The well-characterized mutation of the di-leucine sorting signal in the context of the Nef^LAI^ protein was also tested as a control. Error bars represent s.d. from three biological replicates, an unpaired two-tailed *t*-test, *** *p* > 0.0001 (Graphpad) –(**B**): representative confocal micrographs and quantification of Lck retargeting in Jurkat (TAg) T cells co-expressing the indicated Nef variants and mutants as a fusion protein with eGFP and Lck as a fusion protein with RFP. The quantification of total cells shows a dense accumulation of Lck at the *trans*-Golgi network when counting at least 100 eGFP-expressing cells per condition (plotted as %). Nef^SF2^ and the corresponding loss-of-function mutant AxxA were used as controls. Scale bar: 10 µm. Bars represent the mean ± s.d. (biological triplicates), unpaired two-tailed *t*-test, *** *p* < 0.001, ** *p* < 0.01 (GraphPad).

**Figure 6 viruses-15-00652-f006:**
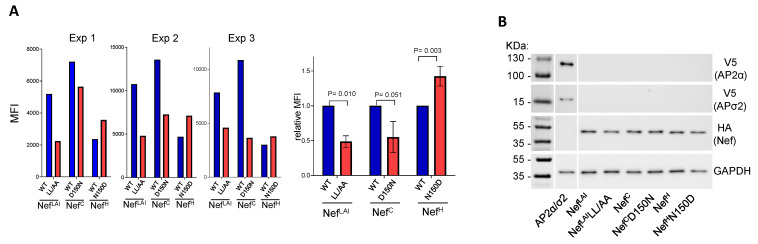
An Asp at Nef residue 150 associates with the higher recruitment of AP2. (**A**): Mean fluorescence intensity (MFI) of BiFC signal from three independent experiments in cells expressing the indicated Nef proteins tagged with Venus N-HA and AP2α/σ2 hemicomplex tagged with V5-Venus C-terminal fragment. Mutation D150N reduced complementation fluorescence from Nef^C^: AP2α/σ2, while substitution N150D in Nef^H^ increased the signal associated with AP2α/σ2 hemicomplex. As a control, the abrogation of AP2 binding motif in Nef^LAI^ (LL/AA) also reduced the association with the AP2α/σ2 hemicomplex, which produced no signal when expressed alone. MFI was measured in three independent experiments (**left**), and presented as average fold BiFC signal over control (WT) Nef (**right**). Mean ± s.d. Shown is the P-value derived from one-sample *t*-test (GraphPad) (**B**). A western blot analysis of AP2α/σ2-VN-V5 and Nef-HA-VC shows no difference in cellular expression between variants.

## Data Availability

All data generated and analyzed in this study are included in this article. Additional data are available upon request from the corresponding author.

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
