# Peer review of "A Conserved Acidic Residue in the C-Terminal Flexible Loop of HIV-1 Nef Contributes to the Activity of SERINC5 and CD4 Downregulation"

_viruses, 2023, doi:10.3390/v15030652_

Round 1
Reviewer 1 Report
Nef is an HIV-1 pathogenic factor contributing to disease progression and a potential target for pharmacological intervention. Nef counteracts the activity of Serinc 5, a host viral restriction factor that incorporates into nascent viral particles making them less infectious. Unlike other important Nef targets (CD4 and MHC-I), information on the structural basis for Serinc 5 antagonism is mostly missing. This study by Firrito and colleagues contributes novel information on Nef molecular determinantes for Serinc5 downregulation and HIV infectivity.
The authors identified a clade H Nef allele that does not counteract Serinc 5-mediated reduction of HIV infectivity. Using functional domain swap experiments, the authors mapped this defect to an asparagine residue in position 150, which replaces an acidic residue (an aspartic or a glutamic acid residue) in Nef alleles that are active against Serinc 5. Consistently, the N150D substitution in the defective Nef reconstituted its ability to downregulate Serinc5 and promote HIV-1 infectivity. The N150D substitution was also sufficient to enable the defective Nef to downregulate CD4, corroborating data indicating that Nef downregulates Seric 5 and CD4 via a shared mechanism (namely AP-2/clathrin-mediated endocytosis). In fact, by bimolecular fluorescence complementation, the authors show that the D150 residue contributes to Nef association with AP-2.
Surprisingly, the reciprocal D150N conversion in the highly active Nef did not strongly affect Serin5 downregulation. The authors discuss that other structural features may compensate for the lack of this otherwise critical acidic residue.
The paper is very clearly written and addresses and interesting issue. My only concern is the lack of information on the number of biological replicates performed for many of the experiments, allowing proper statistical data analysis to support the work's conclusions adequately.
Major concerns
- The infectivity assays shown in Figure 1a seem to have been performed only once, as it is stated: "Error bars represent SD from four technical replicates." This weakens the data and raises questions about the significance of the findings. Information on the number of biological replicates relative to Figure 2 and 4 are also missing.
- The FACS histograms shown in Figures 1, 5, and 6 are representative of how many independent experiments? This information is not available. Also, there are no numeric values in the y or x-axis of the histograms. In addition, an explanation of how the data was normalized to cell counts in each case should also be included.
In Figure 1c, please explain how the "Nef fs" samples were obtained. Are those cells transfected with Serinc5 plus ires2-eGFP plasmids and GFP-gated? This information should also be included in Figure 5.
Minor points
Line 299-303 I found the statement explaining the results in Figure 3e confusing. Are the bars inverted? The data does not show that Nef-CHC is two-fold weaker than Nef C. Also, the data does show that the activity of Nef-HCH hybrid construct is stronger than Nef H, but I still seem weaker than Nef C (not a full rescue). Also, notice that the experiment measured Nef''s effect on viral infectivity and does not directly monitor the "activity of Nef against SERINC5", as stated.
- A representative microscopy image of LCK.RFP distribution under control conditions should be included in Figure 5D.
- Line 107: The "the" word is duplicated;
- Line 108: The word "diffent" is misspelled
- Figure 4E: To remove the "D" that is in the first (white) bar and place it into the third bar;
- Figure 6C: There are two "b" panels.
Reviewer 2 Report
In this manuscript, Firrito et al examine the reason why a type H Nef has lost its ability to counteract SERINC5, rendering HIV-1 sensitive to SERINC5- mediated restriction. Using a series of chimeric Nef molecules, they identified an Asn residue at the base of the C-terminal loop responsible for the attenuation of the anti-SERINC5 function of the type H Nef. The authors found that conversion of this Asn to Asp rendered Nef restrictive towards SERINC5. Overall, this manuscript is well thought out, well written with only some edits needed for clarity. The authors have addressed the questions they have raised in an astute manner. We provide comments on some relatively minor things we identified in the manuscript.
1. The authors in the materials and methods section should provide the number of cells they are transfecting in their experiments.
2. In the Plasmids section of the Materials and Methods, the bibliography formatting does not match the rest of the paper.
3. In line 122-123, the authors say NIH Research and Reagent Reference Reagent Program. This is incorrect, it should be NIH AIDS Reagent Program.
4. In line 164 “…IgG was used, diluted 1:500”. This sentence is unclear. Please rewrite.
5. In line 243, the authors state clade F (HIV90CF056). But in Figure 1A, this is mentioned as Clade H. Which one is it?
6. In Figure 1C, what does the FS stand for in Neffs? The authors need to provide that info in the figure legend.
7. In line 281, the authors refer to Figure 2B and talk about “stability” of the proteins. The authors need to remove that. Figure 2B just shows steady state expression of these proteins. To talk about protein stability, the authors need to do time course experiments involving cyclohexamide treatment assays.
8. In figure 3a, each bracket encloses 23aa, line 287 talks about “20 amino acids” and yet the figure legend of figure 3 and lines 288-304 talk about 22 amino acids. So, what is really the number of amino acids per chimera swapped?
9. In figure 3C, CH130 should be all white and not with some gray at the N’terminus. This reviewer’s understanding is that in CH130, all residues from 130 to 207 are from H.
10. In figure 3D and E, a similar problem as before. The authors label lanes 3 and 4, CHC134-154 and HCH134-154, but in D it is CHC130-152 and HCH130-152. In the methods in lines 101-102, it says CHC131-151 and HCH131-151. In figure 4B it is 134-154. Which one is it?
11. Lines 300-302, the authors state “while the replacement of 22 amino acids of Clade C Nef….the activity of NefH”. This statement does not reflect the data or there is something wrong with the labeling in Figure 2E. Inserting the 22 residues from H to C, was similar to what was seen in C. The effect is seen when the 22 residues of C are added to H. The authors need to rewrite this part or relabel Fig. 3E.
12. Moreover, the CHC130-153 in Fig 3E is really not that different from C, why does the 22 aa fragment from H not result to loss of infectivity as it would be expected? The authors should address it.
13. In line 308, the authors identify three sites to be different. When looking at Fig 3A, 152 position is also different (S to Q). Is there a reason, why this difference was not investigated?
14. Are these amino acid differences (aa134, 136, 150) different in the other Nefs examined in figure 1A? If yes, that should be mentioned in the manuscript.
15. In line 342, the authors refer to the residues Asp and Asn, then later on in the document they are referred with their full name. The authors need to select a consistent nomenclature.
16. In figure 4E, C D150N reduced infectivity to 50%, but the effect on SERINC5 surface levels is not existent. Why is that? The authors need to discuss this, as this reviewer would expect surface levels to go up a bit more than what it is.
17. In Figure 3E, why does C (the third horizontal column) not have its residue indicated like the other 3 columns?
18. In figure 5A, why are the authors now use LAI Nef? It is not really explained in the document.
19. Molecular weights are missing from most of the blots, they should be provided.
Reviewer 3 Report
In this paper, Firrito et al studied the role of a key Nef residue in the downregulation of SERINC5 and CD4.
The paper is rather well focused and easy to read. Approaches were generally well chosen. There are nevertheless some limitations.
Throughout the paper, the authors monitor the expression level of their Nef constructs using lysates of cells transfected with pNL4.3 DEnv DNef + Env vector + Nef vector. An important issue is Nef expression level compared to viral Nef (from WT pNL4.3): what is the extent of overexpression? Please provide an estimate.
In Fig.1a, statistical analysis should be presented.
In Fig.1c, it should be indicated whether a linear or log scale was used for SERINC5 levels.
In Fig.1d it should be indicated in the legend that a logarithmic fit was used.
Regarding Fig.2 where do these cuts (after res65 and res130) fall with respect to Nef structure?
Fig.2a, the EE motif (154-155) that binds b-COP (see PMID: 18725938) should be indicated.
Fig.3e, there is a discrepancy between the text and the Figure regarding the swapped mutants. Please correct.
Fig.5A/B/C, the legend (color code for orange and grey) is lacking.
In Fig.5D the control image (no Nef, just Lck-RFP) should be shown, so that Nef-induced Lck relocalization can be visualized.
Fig.6. There are two b panels. The contrast of the AP2a2 band should be increased.
Regarding the discussion, as mentioned above, the ExxxLL Nef motif (res 155-160 enabling AP-2 binding) overlaps with another motif, the EE motif (154-155) that binds b-COP (PMID: 18725938). The authors should thus broaden their discussion and present the possibility that res150 modulates the function of both motifs, thereby affecting CD4 degradation by Nef both at the level of CD4 endocytosis (Nef-AP2) and targeting to lysosomes (Nef-b-COP).
In the methods please identify the kit used for site-directed mutagenesis.
